# The Effect of Rainfall and Temperature Patterns on Childhood Linear Growth in the Tropics: Systematic Review and Meta-Analysis

**DOI:** 10.3390/ijerph21101269

**Published:** 2024-09-25

**Authors:** Derese Tamiru Desta, Tadesse Fikre Teferra, Samson Gebremedhin

**Affiliations:** 1School of Nutrition, Food Science and Technology, Hawassa University, Hawassa P.O. Box 05, Ethiopia; tadessefikre@hu.edu.et; 2Institute for Enhancing Health through Agriculture, IHA, Texas A&M University, College Station, TX 77843, USA; 3School of Public Health, Addis Ababa University, Addis Ababa P.O. Box 12485, Ethiopia; samsongmgs@yahoo.com

**Keywords:** temperature, rainfall, stunting, tropics, under-five, weather

## Abstract

Despite existing research on child undernutrition in the tropics, a comprehensive understanding of how weather patterns impact childhood growth remains limited. This study summarizes and estimates the effect of rainfall and temperature patterns on childhood linear growth among under-fives in the tropics. A total of 41 out of 829 studies were considered based on preset inclusion criteria. Standardized regression coefficients (β) were used to estimate effect sizes, which were subsequently pooled, and forest plots were generated to visually represent the effect size estimates along with their 95% confidence intervals. Of the total reports, 28 and 13 research articles were included in the narrative synthesis and meta-analysis, respectively. The studies establish that patterns in rainfall and temperature either increase or decrease childhood linear growth and the risk of stunting. An increase in every one standard deviation of rainfall results in a 0.049 standard deviation increase in linear growth (β = 0.049, 95% CI: 0.024 to 0.073). This positive association is likely mediated by various factors. In countries where agriculture is heavily dependent on rainfall, increased precipitation can lead to higher crop yields which could in turn result in improved food security. The improved food security positively impacts childhood nutrition and growth. However, the extent to which these benefits are realized can vary depending on moderating factors such as location and socio-economic status. Temperature pattern showed a negative correlation with linear growth, where each standard deviation increase resulted in a decrease in linear growth by 0.039 standard deviations, with specific impacts varying by regional climates (β = −0.039, 95% CI: −0.065 to −0.013). Additionally, our meta-analysis shows a small but positive relationship of childhood stunting with temperature pattern in western Africa (β = 0.064, 95% CI: 0.035, 0.093). This association is likely due to temperature patterns’ indirect effects on food security and increased disease burden. Thus, the intricate interaction between weather patterns and childhood linear growth requires further research to distinguish the relationship considering other factors in the global tropics. While our findings provide valuable insights, they are primarily based on observational studies from sub-Saharan Africa and may not be generalizable to other tropical regions.

## 1. Introduction

Climate-change-induced extreme weather events are becoming more common and intense, endangering global food security, nutrition, and public health [1]. The events encompass a range of phenomena, including heavy precipitation, increasing temperatures, prolonged droughts, devastating floods, and heat waves. Research highlights the complex and multifaceted threats posed by these extreme weather events to child health and nutrition globally, with a particular focus on the tropics. The trend is noticeably accelerating with adverse effects, and projections are indicating a substantial rise, as reported by the Lancet Countdown report [2]. Countries within the global tropics are particularly vulnerable to climate-change-induced weather events [3]. The 2020 State of the Tropics Report underscores a worrying situation [4]. Due to the increased frequency of extreme weather occurrences, fragile ecological communities, limited adaptive capacity, and reliance on agriculture, the global tropics stand as a region particularly vulnerable to the adverse effects of climate change [5].

The impact of rising temperatures and rainfall variations on child growth remains a complex topic with ongoing research. Figure 1 depicts the complex interplay of rainfall and temperature as a result of climate change with childhood linear growth, highlighting geographic characteristics and moderating and mediating factors at play. Geographic factors, including latitude and altitude, also affect global weather conditions, including temperature and rainfall patterns. Areas located near the tropics remain warm and relatively wet throughout the year [6]. However, the climatic changes induce rainfall and temperature variations, which directly impact childhood linear growth, moderated by several socio-economic factors. The moderating factors include socio-economic status, which may encompass poverty, education, and livelihoods, which in turn influence the resilience and adaptability of communities to climate impacts. The burden on women is particularly high. Women in developing countries who play a central role in agricultural activities face a compounding burden. Women are responsible for essential household tasks like water collection, physically demanding activity further strained by seasonal variations and drought. During the rainy season, increased agricultural work demands more of their time. Droughts necessitate traveling longer distances for water and firewood, adding significant stress [7].

The mediating factors further elucidate the pathways through which climate change and weather conditions affect childhood linear growth. Countries reliant on subsistence agriculture are particularly susceptible to these shocks, experiencing an increased risk of child undernutrition [8]. Food and nutrition security is compromised through the decreased availability of nutritious foods [9], decreased crop nutrient density, food spoilage, and limited access to clean water [10]. Multiple studies [11,12] have documented declining crop production in tropical regions due to extreme weather events. On the other hand, research suggests compromised nutrient density in crops as a result of elevated temperatures and CO2 levels [13,14]. Meanwhile, indirectly, climate-change-induced extremes impact human well-being through deteriorating food security and livelihoods. On top of these, environmental nutrient cycling is also impacted by climate change. This will, in turn, affect soil carbon and plant growth, reducing food production [15]. This would result in “hidden hunger” where calorie intake does not meet nutritional needs, disproportionately affecting populations reliant on susceptible crops like rice [16]. Therefore, climate-change-induced temperature and rainfall patterns primarily affect child linear growth by influencing agricultural production. Particularly extreme weather events like heatwaves [17] and droughts [18] can directly harm crops, causing food shortages and limiting children’s access to nutritious food.

Low rainfall (droughts) and heat waves exacerbate chronic undernutrition in children through diminished food production, leading to food insecurity and an increased incidence of diseases often associated with such events [19,20]. Increased risk of infections such as malaria and diarrhea, along with obstacles to sanitation and healthcare access, exacerbates the vulnerability of children [19]. Additionally, physiological effects like heat stress and dehydration due to drought directly influence childhood growth and health [21]. In general, linear growth in children is not determined by a single factor but rather by the complex interaction of various influences.

Previous reviews have focused broadly on undernutrition without differentiating impacts in the tropical countries. A review conducted in low- and middle-income countries showed that weather variables such as rainfall, extreme weather events (floods and droughts), seasonality, and temperature are associated with childhood stunting at the household level. The study suggested that agricultural, socio-economic, and demographic factors at the household and individual levels also play substantial roles in mediating the nutritional impacts [19]. Another review linked climate change proxies with malnutrition in both children and adults, suggesting a significant relationship between climate change proxies and at least one malnutrition metric [22]. However, none of the reviews focused specifically on childhood linear growth or stunting in the global tropics. Hence, despite existing research on child undernutrition in tropical regions, a comprehensive understanding of how temperature and rainfall patterns impact child linear growth remains limited. Our study contributes to the literature by homing in on under-fives, who are at a critical developmental stage. We examine linear growth as the primary outcome and stunting as a secondary outcome of interest in under-five children. By generating robust evidence on these crucial relationships, the study seeks to inform and support evidence-based decision-making, ultimately contributing to improved child health outcomes.

## 2. Materials and Methods

### 2.1. Searching Strategies

Literature searches were conducted using various research databases, such as EBSCO, MEDLINE through PubMed, EMBASE, Science Direct, Scopus, Mednar, Worldwide Science, and Google Scholar. Journal articles published in English from July 2000 to July 2024 were included. During searching, terms and key words were alternatively combined using the Boolean operators (AND, OR, NOT). The key search terms in combination were [“climate variability” or “weather” or “rainfall” or “temperature” or “precipitation” or “season*” or “drought” or “flood”] AND [“stunting” or “growth disorder” or “undernutrition” or “growth disorder” or “height for age z score” or “malnutrition assessment” or “length for age” or “length-for-age” or “haz” or “short*”] AND name of each country in the tropical regions. All countries in the full tropical region of 23.27° North and 23.27° South [23] were included in the search engine. Growth disorders as a MESH term for stunting were searched in PubMed specifically within the peer-reviewed journal databases. Grey literature searches using Google Scholar and International Food Policy Research Institute were also performed to ensure that other unpublished research outputs were included. All search terms and a list of the countries included in the search engines are listed in Appendix A.

### 2.2. Eligibility Criteria

Inclusion criteria: The systematic review and meta-analysis focused on observational studies investigating both child linear growth and stunting as defined by the WHO growth standards [24]. Participants were children aged between 0 and 59 months in the full tropical regions, encompassing latitudes between 23.27° N and 23.27° S worldwide. Observational study designs, including cross-sectional, cohort, case-control, and national surveys that reported original data on the association between the weather metrics and childhood linear growth, were considered. To ensure comprehensiveness and recent findings, studies published between 2000 and 2024 were considered for inclusion. Furthermore, only studies published in English were included.

Exclusion criteria: This study employed predefined criteria to ensure the quality and relevance of the included publications. Studies were excluded if they lacked an abstract or full text, as these elements are crucial for evaluating their eligibility based on the outlined criteria. Additionally, anonymous reports were excluded due to the absence of information about authorship and the potential for bias. Editorials and commentaries were not considered as they do not present primary research data relevant to this analysis. Systematic reviews and meta-analyses were also excluded, as they are not the primary focus of this study and their methodologies overlap with the ones employed here. Finally, qualitative studies were excluded because this review focuses on quantitative data in the meta-analysis. The methods used in qualitative research differ significantly from those employed in quantitative studies, thus making them unsuitable for meta-analysis.

### 2.3. Study Variables

This study investigates the effect of rainfall and temperature patterns on linear growth (height-for-age z-score) and child linear growth faltering (stunting), defined by the World Health Organization (WHO) as a height-for-age z-score below −2 standard deviation (SD) among children under five [25] in the global tropics. Weather proxy variables, including rainfall and temperature, were the independent variables. Rainfall and temperature serve as indispensable proxy indicators for weather patterns due to their profound influence on ecosystems, human endeavors, and the overarching climate system. Through monitoring and analysis of these parameters, researchers can acquire invaluable insights into the intricacies of climate change, its attendant impacts, and potential future trajectories [26]. To comprehensively assess the impact of rainfall pattern, this study incorporated a broad range of studies examining patterns in rainfall. This included studies focusing on changes in rainfall amount and timing, and broader measures encompassing changes in precipitation patterns. Additionally, research investigating aridity, a direct indicator of reduced rainfall, was included. The temperature pattern was similarly defined to encompass a range of indicators used in the original study. Studies were included if they referenced increases in temperature, arid conditions, or average temperatures. Additionally, studies examining the impact of early-life exposure to anomalous temperature conditions, such as a 10% increase in days below 15 °C, were incorporated into this definition.

### 2.4. Quality Assessment

The quality assessment was conducted in two stages. Initially, one reviewer (DTD) made a quality assessment. In the second stage, two reviewers (DTD and SG) checked whether the presumed quality had been maintained. Critical appraisal of methodological quality of all included studies was conducted using the Joanna Briggs Institute’s (JBI) Critical Appraisal Checklist for observational studies. Quality criteria employed in the assessment of included studies encompassed sample size and characteristics, clear study objectives, identification of confounding factors, strategies for controlling confounders, the validity and reliability of outcome measurement, and the statistical analysis techniques utilized. Regression models were predominantly employed across the studies included in the meta-analysis). The models effectively accounted for potential confounders such as child characteristics, parental characteristics, household characteristics, and environmental factors. They were controlled either by explicitly incorporating them as independent variables or by addressing unobserved variation at different levels of the data.

Studies were categorized as having a “high”, “medium”, or “low risk of bias” based on their adherence to the established criteria. A cut-off of 50% or above on the JBI checklist was used to designate studies as “low risk of bias” [27]. Two reviewers agreed upon quality ratings of “low risk of bias” and “high risk of bias” (DTD and SG). All the studies assessed are listed in Appendix A. Consequently, the conclusions drawn from the meta-analysis have been carefully checked in light of the potential effects of confounding factors.

### 2.5. Data Extraction

The data extraction was conducted in two stages. First, the extraction was conducted by DTD and verified by another reviewer (SG). Finally, all the extracted data were verified independently by two reviewers (DTD and SG). The extraction adhered to predefined eligibility criteria. Following a standardized format within Microsoft Excel, data were extracted from each original research article, including the name of the first author, publication year, country/region, study design, sample size, weather metrics, participant age, dependent variable, model employed, and effect sizes. The majority of selected studies were population-based, cross-sectional designs, with nearly all utilizing nationally representative surveys. Notably, only one dynamic cohort and one longitudinal study were included. Eleven studies reported the effect size using standardized regression coefficients along with their standard error. In this case, both the coefficient and standard errors were extracted directly. However, there were studies that reported only the standardized regression coefficients without the standard error, but with *p*-values. In this case, the standard error was computed by dividing the standardized regression coefficient by the confidence level of the regression coefficient. The reported *p*-values in the selected studies were taken at their upper limit [28].

### 2.6. Grading the Evidence

The overall confidence in the evidence was assessed using the revised Grading of Recommendations, Assessment, Development, and Evaluations (GRADE) methodology. Two independent reviewers (DTD and TFT) conducted the GRADE assessments.

### 2.7. Data Synthesis and Analysis

In this systematic review and meta-analysis, the studies included used different effect size measures, including odds ratios and standardized regression coefficients from different regression models measuring the relationship of rainfall and temperature as weather conditions with child linear growth (height-for-age z-scores) and stunting. We used standardized regression coefficients (β) as effect size estimates and reported them in this systematic review and meta-analysis to show the effect of rainfall and temperature variations on child linear growth and stunting. A forest plot was used to visually represent the effect size estimates and their 95% confidence intervals across studies. I-squared statistics were calculated to quantify the heterogeneity, with values of 25%, 50%, and 75% interpreted as low, moderate, and high heterogeneity, respectively [29]. The pooled effect sizes of the regression coefficient (β) were interpreted as small if the effect size was 0.1–0.29, medium if the effect size was 0.30–0.49, and large if the effect size was ≥0.50 [30]. Stata 16 statistical software was utilized for the meta-analysis.

Sensitivity analysis was conducted to select a model and explore the potential influence of outliers (studies with very different results) on the overall findings. This involved leave-one-out analysis, re-running the analysis after excluding individual studies, and observing changes in the pooled estimates [31]. Acknowledging the substantial heterogeneity in effect sizes across the included studies, we opted for a random-effects model selection process. Two prominent models, DerSimonian–Laird (DL) and Sidik–Jonkman (SJ), were considered. Ultimately, the SJ method was chosen for the final reported results based on the application of a distinct weighting scheme specifically designed to support robustness in circumstances that are characterized by high between-study variance in the effect sizes. The SJ method is generally recognized as a potentially superior alternative to the DL method in meta-analysis contexts, particularly when encountering substantial heterogeneity or a limited number of studies included in the analysis [32]. A funnel plot was used to report publication bias [33]. Standard and contour-enhanced funnel plots were used to visualize the publication bias (Appendix A). Additionally, regression-based Egger tests and trim-and-fill analysis were conducted and reported in the Section 3 of the study.

### 2.8. Registration and Reporting

The current systematic review and meta-analysis adhered to the rigorous standards of the Preferred Reporting Items for Systematic Reviews and Meta-Analyses (PRISMA) guidelines [34]. Transparency and reproducibility were ensured, and the review protocol has been registered with the International Prospective Register of Systematic Reviews (PROSPERO) under the registration ID of CRD42024536742.

## 3. Results

### 3.1. Study Selection and Characteristics

In this systematic review and meta-analysis, a total of 42 out of 829 studies fulfilled the inclusion criteria (Figure 2). Of the total, 28 studies were included in the narrative synthesis and 14 studies in meta-analysis. Table 1 and Table 2 show the summary characteristics of studies included in the systematic review and meta-analysis. A total of 158,987 children under five (73,922 for rainfall pattern and 102,901 for temperature pattern) were included in the meta-analysis. The narrative synthesis included 22 studies investigating the relationship between rainfall pattern (encompassing both rainfall and precipitation) and childhood linear growth, as measured by height-for-age z-score (HAZ). Conversely, only two studies explored the association of temperature pattern with HAZ, and eleven with stunting. Among the studies included in the meta-analysis, six examined the impact of rainfall on childhood linear growth, as assessed by height-for-age z-score (HAZ). In contrast, only four studies investigated the association of temperature pattern with HAZ, while another four studies focused on its link to stunting.

Geographically, the majority of the studies were from sub-Sahara Africa (*n* = 15), with Ethiopia and Burkina Faso the most represented (*n* = 4 each), followed by Uganda (*n* = 3), and a single study each from Rwanda, Nigeria, and Sierra Leone. From Asia, Indonesia was included, and from South America, Peru (Figure 3 and Figure 4).

### 3.2. Narrative Synthesis

The effect of rainfall and temperature patterns on childhood linear growth in the global tropics is summarized in Table 1. The studies covered geographically diverse countries, including Uganda, Kenya, Ethiopia, Somalia, Burundi, Rwanda, Tanzania, and Malawi from the eastern part of Africa; Ghana, Nigeria, Burkina Faso, Guinea, Mali, Niger, and Senegal from the western part of Africa; Zimbabwe from southern Africa; and Indonesia and Cambodia from southeastern Asia (Figure 3 and Figure 4). Among the twenty-eight studies included, six examined the effect of rainfall and temperature patterns on childhood linear growth. Twenty reports studied the effect of rainfall and temperature patterns on childhood stunting (defined as height-for-age z-scores below −2 SD). Overall, 22 studies reported statistically significant associations between rainfall and temperature patterns with childhood linear growth. These associations are complex and influenced by various factors, including timing and amount of rainfall, location-specific disease prevalence, conflict, and agricultural practices. While rainfall and temperature are crucial factors, their impact on child stunting appears multifaceted and found with conflicting reports. However, four studies did not find statistically significant associations.

#### 3.2.1. Rainfall Pattern and Childhood Linear Growth

The results of the current analysis have yielded heterogeneous results, with some demonstrating positive associations between rainfall and growth, while others report no significant association or even detrimental effects. All studies reporting a positive association between rainfall and child growth or stunting are from Africa. Six studies originated in East Africa (Ethiopia: 2, Uganda: 2, Kenya: 1, Rwanda: 1) and one in Southeast Africa (Malawi), while three were conducted in West Africa (Nigeria: 2, Burkina Faso: 1).

In Ethiopia, within administrative zones, one standard deviation increase in rainfall was linked to a 0.242 standard deviation rise in moderate stunting prevalence [35]. Similarly, a positive association between early life kiremt (rainy season) rainfall and child height-for-age z-scores, with a one-centimeter increment in rainfall linked to a 0.012 unit increase in height-for-age z-scores, was reported [7]. Within the central and eastern regions of Uganda, a statistically significant increase (*p* < 0.05) of stunting prevalence due to increased mean rainfall was reported [36]. In Nigeria, children residing in areas with moderate rainfall (142–1199 mm annually) were less likely to experience stunting compared to those in low-rainfall areas (odds ratio = 0.78; 95% credible interval [CI]: 0.64, 0.96) [37]. Additionally, a positive association (β = 0.007) was reported in a rural setting [38]. Another study found a statistically significant positive association between higher cumulative rainfall over extended periods (36 months) and height-for-age z-scores in Ghana [39]. Similarly in Burkina Faso, a strong and positive association was reported (β = 0.481) [40].

Despite the positive associations, a study in Indonesia found no statistically significant independent association between early childhood precipitation exposure and height-for-age z-scores [41]. A study done in Uganda found no significant effect of increased rainfall patterns on chronic undernutrition (stunting) [42]. Again, in Uganda, a study reported no statistically significant association between annual rainfall exceeding long-term averages (positive deviations) and reduced stunting rates [43].

However, a study done in Lake Victoria Basin countries showed an increase in child stunting rates due to increased rainfall [44], while in Malawi, shorter seasonal rainfall durations and below-average seasonal rainfall increased the prevalence of stunting [45]. Another study in Malawi also showed that rainfall had a positive effect on stunting (β = 0.076, *p* = 0.044) [46]. On the other hand, a contradictory finding was reported, indicating higher stunting prevalence associated with residents in both less arid and increased-precipitation regions, particularly in western Ghana [47].

Apart from the association, the timing and location of rainfall and the social context appear to influence the relationships. In Indonesia, there was a differential impact of early-life rainfall on child health, with a positive association between rainfall in the first 1–3 months of life and nutritional outcomes in rural areas. The study demonstrated that rainfall in the first 1–3 months of life is associated with a higher height potential (0.13 point increase in z-score) for a child experiencing average monthly rainfall (200 mm) compared to no rain [48]. In Somalia, two studies [49,50] showed that decreased rainfall was a significant factor associated with an increased risk of stunting (OR = 0.994, 95% CI: 0.993, 0.995). In particular, after adjusting for conflict, increased rainfall had a statistically significant protective effect on stunting (OR = 0.86, 95% CI: 0.85–0.87). This suggests that adequate rainfall may mitigate undernutrition, but its impact is likely masked by social factors.

#### 3.2.2. Temperature Pattern and Childhood Linear Growth

Seven studies investigated the effect of temperature on child linear growth and stunting in Africa (Ethiopia: 2, Tanzania: 2, Nigeria: 1, Burkina Faso: 1). The studies reported both positive and negative associations, considering that other factors affecting child nutrition. In Ethiopia, it was observed that a one-unit increase in temperature is linked to a 0.216 standard deviation decrease in moderate stunting prevalence [35]. Another similar study found a 0.19 standard deviation decrease in stunting prevalence with a one-unit increase in temperature [51]. A contrasting pattern in Nigeria was reported, where a statistically significant positive association emerged, with a one-degree Celsius increase in temperature translating to a 16.7% rise in the probability of stunting [52].

The potential for geographically specific effects is highlighted by another study [53]. The study found contrasting patterns in northern and southern Mali. While higher average temperatures over two years were associated with increased stunting risk in the north, the opposite trend emerged in the south, suggesting a potential moderating effect of location. Additionally, prenatal temperature exposure appears to play a crucial role. In utero exposure to high temperatures (exceeding 29 °C) in Tanzania was linked to lower postnatal height-for-age z-scores in boys, suggesting a potential sex-specific vulnerability [54]. Another study [55] further emphasized the critical nature of the second trimester in Tanzania, with colder-than-usual temperatures during this period linked to an increased risk of stunting.

#### 3.2.3. Combined Effects of Rainfall and Temperature Patterns on Child Linear Growth and Stunting

Four studies investigating the combined effects of rainfall and temperature on child growth and stunting report heterogeneous results. The findings highlight the need for a geographically specific understanding of how environmental factors interact to influence child health outcomes. A negligible association between these factors and childhood malnutrition in Cambodia was reported [56]. Conversely, in Ethiopia, a significantly higher likelihood of stunting among children residing in arid regions with lower rainfall and higher temperatures (OR = 0.83, 95% CrI: 0.70, 0.999) was observed [57].

A complex relationship between climate and stunting in Ghana was observed. While high aridity (low rainfall and high temperatures) was associated with decreased stunting prevalence, increases in precipitation were linked to a rise in stunting [58]. In contrast, a study conducted in a multi-country analysis across sub-Saharan Africa found a positive association between rising temperatures and low rainfall (drying conditions) and increased rates of stunting [59].

**Table 1 ijerph-21-01269-t001:** Summary of the effects of temperature and rainfall patterns on child growth in the full global tropics, 2024.

Exposure	Effect on Child Linear Growth or Stunting	Country/Region
Mean rainfall	Exacerbates childhood stunting disparities across districts	Uganda [36]
Higher rainfall	Contributes to alleviating food insecurity but may paradoxically elevate undernutrition, including stunting	Ghana [39]
Increase in rainfall	Improves linear growth (height-for-age z-scores)	Ethiopia [7]
Rainfall in the first 1–3 months of life	Associated with higher height potential (0.13-point increase in z-score) for a child experiencing average monthly rainfall (200mm) compared to no rain	Indonesia [48]
Residing in a medium-rainfall geographic area (142–1199 mm rainfall)	Is positively associated with an increased prevalence of stunting	Nigeria [37]
Residing in less arid areas (i.e., areas with more rain)	Potentially contributes to increased stunting prevalence	Ghana [47]
Increases in precipitation	Coincides with an increase in stunting prevalence, particularly in western Ghana	
Poor rainfall	Results in undernutrition (stunting)	Somalia [49,60]
Increases in rainfall variability	Shows no significant association with chronic undernutrition (stunting)	Uganda [42]
The level of precipitation	Significantly predicts higher height-for-age z-scores	Kenya [61]
Increase in rainfall	Increases rate of child stunting	Lake Victoria Basin countries including Burundi, Kenya, Rwanda, Tanzania, and Uganda [44]
Positive annual deviations (greater rainfall) from long-term precipitation trends	Shows no significant association with chronic undernutrition (stunting)	Uganda [43]
Season rainfall duration and below-average seasonal rainfall	Increases stunting	Malawi [45]
Early childhood precipitation exposures	Not independently associated with height-for-age z-score	Indonesia [41]
Increase in rainfall	Increase in moderate stunting	Ethiopia [35]
Increase in temperature	Decrease in moderate stunting	
Rainfall and temperature	Has negligible impacts on malnutrition, underscoring the multifaceted nature of environmental influences on child health	Cambodia [56]
Residing in arid geographical locations (characterized by lower rainfall and higher temperature)	Increases the likelihood of stunting	Ethiopia [57]
High aridity characterized by low rainfall and higher temperature	Decreases stunting prevalence (negative association)	Ghana [58]
Increases in precipitation	Contributes to a rise in the prevalence of stunting	
Temperature and rainfall variability	Causes retarded linear growth with increasing incidences of disease	Tanzania [54]
Increases in warming and drying	Leads to a rise in the incidence of linear growth faltering (stunting)	Countries in sub-Saharan Africa including Ethiopia, Kenya, Madagascar, Malawi, Rwanda, Uganda, Zimbabwe, Burkina Faso, Guinea, Mali, Niger, Nigeria, and Senegal [59]
Increase in temperature	Contributes to a reduction in the incidence of stunting	Ethiopia [51]
Higher temperature	Results in a higher length-for-age z-score	Burkina Faso [62]
Increased average temperatures	Exacerbates the susceptibility of children to linear growth faltering (stunting)	Northern Mali [53]
Rise in temperature	Increases incidence of child stunting	Nigeria [52]
One standard deviation from the long-term mean	Increases the likelihood of stunting and severe stunting	Tanzania [55]
High rainfall	Is associated with a 1.58-fold increased risk of stunting	Rwanda [63]
Rainfall	Is positively associated with childhood linear growth	Nigeria [38] and Burkina Faso [40]
Rainfall	Is positively associated with stunting	Malawi [46]

### 3.3. Meta-Analysis

In the meta-analysis, the effect of rainfall patterns on height-for-age z-score (HAZ) was assessed by pooling regression coefficients from six studies. On the other hand, the analysis of temperature patterns employed a more nuanced approach due to the heterogeneity in temperature pattern assessment. Four regression coefficients were reported on the effects of extreme temperatures (a single study contributed a regression coefficient for high temperatures > 26 °C and low temperatures < 16 °C). Thus, data from two studies reporting on extreme temperatures and a single study investigating the effect of average temperature across various child age categories were pooled to evaluate the overall effect of temperature pattern. Finally, a separate analysis using pooled regression coefficients from four studies examined the association between temperature pattern and stunting. However, a meta-analysis for the effect of rainfall pattern on childhood stunting was not performed due to there being a single report. The studies included in the meta-analysis in Table 2 produced inconsistent findings, with some indicating that changes in rainfall and temperature could either promote or hinder childhood growth and stunting, while others found no link. The conflicting results across studies may reflect underlying variations in regional dietary diversity, agricultural practices, and community resilience to climate impacts.

**Table 2 ijerph-21-01269-t002:** Studies included in the meta-analysis (n = 16), 2024.

Author	Country/Region	Design	Age in Months	Sample Size (n)	Dependent Variable	Weather Metrics	Model Used
Ayalew [64]	Ethiopia	Cross sectional	6–36	17,836	Height-for-age z-scores (HAZ)	Temperature	Fixed-effect cross-section model
Amegbor et al. [36]	Uganda	Cross-sectional	0–59	3625	Stunting	Temperature	Multilevel mixed-effect analysis
Blom et al. [65]	West Africa (Benin, Burkina Faso, Cote d’Ivoire, Ghana and Togo)	Cross-sectional	3–36	32,036	Stunting	Temperature	Ordinary least squares
Injete Amondo et al. [66]	Uganda	Cross-sectional	7–59	4921	Height-for-age z-scores (HAZ)	Temperature	Fixed-effect regression
Randell et al. [7]	Ethiopia	Cross-sectional	12–59	23,026	Stunting	Temperature	Multivariate regression models
Thiede and Gray [41]	Indonesia	Cross-sectional	0–11	7459	Height-for-age z-scores (HAZ)	Temperature	Fixed-effect regression models
Abiona [67]	Rural Sierra Leone	Cross-sectional	0–59	1677	Stunting	Temperature	Fixed-effect models
Rojas et al. [68]	Burkina Faso	Cross-sectional	24–59	12,321	Height-for-age z-scores (HAZ)	Temperature	Fixed-effect regressions
Ayalew [64]	Ethiopia	Cross-sectional	6–36	17,835	Height-for-age z-scores (HAZ)	Rainfall	Fixed-effect cross-sectional model
Nicholas et al. [69]	Peru	Cross-sectional	24–60	13,484	Height-for-age z-scores (HAZ)	Rainfall	Linear models
Randell et al. [7]	Ethiopia	Cross-sectional	12–59	23,026	Height-for-age z-scores (HAZ)	Rainfall	Multivariate regression models
Ssentongo et al. [70]	Uganda	Cross-sectional	0–59	5219	Height-for-age z-scores (HAZ)	Rainfall	Linear regression
Yeboah et al. [71]	Burkina Faso	Cross-sectional	0–59	12,919	Height-for-age z-scores (HAZ)	Rainfall	Multilevel regression
Mank et al. [72]	Burkina Faso	Dynamic cohort	7–60	1439	Height-for-age z-scores (HAZ)	Rainfall	Multilevel regression analysis

#### 3.3.1. Rainfall Pattern and Childhood Linear Growth

A meta-analysis encompassing five cross-sectional studies and one dynamic cohort investigated the association between rainfall pattern and childhood linear growth (height-for-age z-scores) across diverse geographical regions of the tropics (Ethiopia: 2, Burkina Faso: 2, Uganda: 1, Peru: 1). The analysis revealed contrasting findings based on rainfall patterns. Three studies employing metrics of lower rainfall, such as mean average lifetime rainfall exposure [71], precipitation pattern with more consecutive dry days [72], and lower monthly precipitation exposure [64], reported negative associations with childhood linear growth. Conversely, three other studies examining higher rainfall metrics, including higher postnatal rainfall [69], cumulative rainfall from birth to current age, increased annual precipitation (by one standard deviation, approximately 170 mm) [70], and high rainfall exposure from birth to current age [7], documented positive associations with childhood linear growth. These divergent findings suggest potential geographical variations in the influence of rainfall patterns on children’s linear growth.

The forest plot (Figure 5) revealed a non-significant overall effect size (β = −0.01, 95% CI: −0.08, 0.05) for the association between rainfall variations and childhood linear growth. The standardized beta coefficient (β) of −0.01 (*p* = 0.71) suggests a small decrease in childhood linear growth for every one standard deviation increase in rainfall. However, we found a statistically significant Chi-square test result (Q = 144.55, *p* < 0.0001) alongside high heterogeneity (I^2^ = 95.87%) in effect sizes across studies, while the estimated Tau-squared (τ^2^ = 0.01) shows low between-study variance. The values indicate that while the overall between-study variance might be low, there are likely substantial differences between specific subgroups of studies, requiring subgroup analysis [73].

In this regard, the divergent findings from the included studies suggest a potential influence of rainfall patterns on children’s linear growth. On the other hand, two specific patterns showing a geographic clustering of the reported regression coefficients emerged across western and eastern African regions. Studies with smaller sample sizes have lower statistical power, making it less likely to identify significant effects. This can result in a wider range of effect sizes in the meta-analysis, as some studies may find significant effects while others do not [74]. Therefore, considering these factors, the source of heterogeneity was explored using a subgroup analysis categorizing rainfall variations into higher and lower rainfall metrics (Table 3). The analysis revealed contrasting findings. Higher rainfall was linked to positive linear growth (β = 0.049, 95% CI: 0.024 to 0.073), indicating faster growth in children. This implies that childhood linear growth increases by 0.049 standard deviations for every standard deviation increase in rainfall. Conversely, exposure to lower rainfall was associated with negative linear growth (β = −0.080, 95% CI: −0.140 to −0.020). The pooled standardized beta coefficient (β) of −0.08 suggests that lower rainfall exposure is associated with decreased childhood linear growth.

##### Publication Bias

An initial visual inspection of the funnel plot (Appendix A) indicated an absence of overt publication bias in the analysis of rainfall variations and childhood linear growth. To formally assess this, a regression-based Egger test incorporating the Sidik–Jonkman method was conducted within a random-effects framework. The resulting coefficient estimate (β = −1.41, *p* = 0.5047) hinted at a potential negative association between effect size and standard error. However, this trend did not reach statistical significance (*p* > 0.05), suggesting that small-study effects were unlikely to have substantially impacted the overall findings. Furthermore, the trim-and-fill method did not detect any missing studies, and the effect size (β = −0.013, 95% CI: −0.079 to 0.053) remained consistent across both observed and imputed datasets. Collectively, these results provide robust evidence against the presence of significant publication bias in the meta-analysis.

#### 3.3.2. Temperature Pattern and Childhood Linear Growth

Figure 6 presents a forest plot summarizing the findings of this systematic review and meta-analysis regarding the impact of temperature pattern on childhood linear growth within the included tropical countries (1 each from Ethiopia, Indonesia, Uganda, and Burkina Faso). The studies included in the analysis reported heterogeneous results. In Ethiopia, a study [64] showed that high temperature (above 26 °C) was positively associated with child linear growth (β = 0.041; 95% CI: −0.047 to 0.129). In contrast, the studies in Uganda [20] and Burkina Faso [68] reported statistically significant negative effects of frequent heatwaves (defined as at least three occurrences in the past 5 years) on childhood growth. The Ethiopian study [64] also found a non-significant negative association of low temperatures (<16 °C). On the other hand, the Indonesian study [41] reported the influence of mean daily temperature on growth across two age groups (0–11 months and 12–23 months), and there was no statistically significant association between the variables across the age groups. However, in the subgroup analysis, the effect of extreme temperatures demonstrated a statistically significant negative association. The pooled standardized regression coefficient (β) was −0.041, with a 95% CI: −0.069 to −0.014. This indicates that children exposed to more extreme temperatures (both high and low) tend to have slightly lower childhood linear growth on average in tropical countries.

##### Publication Bias

The funnel plot presented in Appendix A demonstrates pronounced asymmetry, with an overrepresentation of studies reporting smaller effect sizes. A distinct paucity of studies is evident in the left region of the plot, particularly those with smaller negative or positive effect estimates. Thus the pattern strongly suggests the potential for publication bias, as unpublished studies with non-significant or negative findings may be underrepresented in the meta-analysis [73].

Furthermore, a statistically significant correlation between effect size and sample size was observed. Smaller studies tended to exhibit more extreme effect estimates compared to larger studies. The discrepancy between observed and imputed effect sizes reinforces the hypothesis of missing studies with smaller, potentially negative or null effects, likely attributable to selective publication of studies with statistically significant outcomes.

#### 3.3.3. Temperature Pattern and Stunting

Four studies were pooled to determine the effect of temperature pattern on childhood linear growth in Africa (multi-country encompassing western African countries Benin, Burkina Faso, Cote d’Ivoire, Ghana, and Togo: 1; Uganda: 1; Ethiopia: 1; and Sierra Leone: 1). Figure 7 presents the effects of temperature patterns on stunting in tropical countries. The overall effect size (β) was −0.003, with a 95% CI: −0.23 to 0.22. This shows that the effect of temperature on childhood linear growth failure (height-for-age z-score < −2 SD) was not statistically significant. The heterogeneity test (I^2^ statistic) shows a significant amount of heterogeneity between the studies (I^2^ = 97%). The test for overall effect (*p*-value = 0.98) shows that the overall effect size is not statistically significant from zero. Thus, this meta-analysis found no statistically significant evidence that both temperature pattern and mean annual temperature exposure have an effect on childhood linear growth failure (height-for-age z-score < −2SD).

##### Subgroup Analysis

Given the significant heterogeneity observed across the studies, we conducted subgroup analyses to explore potential sources of variation, considering factors similar to the rainfall patterns and childhood linear growth. Table 4 shows subgroup analyses considering factors including region and sample size. The subgroup analysis of studies conducted in the West African region and sample size category may have contributed to the overall heterogeneity. Two studies [65,67] investigated the effects of exposure to heat and temperature shocks in West Africa. The meta-analysis of these studies, encompassing data from Benin, Burkina Faso, Cote d’Ivoire, Ghana, and Togo [65], as well as rural Sierra Leone [67], revealed a statistically significant positive pooled effect size (β) of 0.064 (95% CI: 0.035 to 0.093). This indicates that children experiencing greater variability in temperature (deviations from the average) are likely to have a slightly higher risk of stunting (defined as a HAZ score lower than −2 SD) in the West African countries.

##### Publication Bias

The funnel plot depicted in Appendix A exhibits an asymmetrical pattern, indicative of potential publication bias. This suggests a disproportionate representation of studies with statistically significant findings compared to those without. Supporting this impression, Egger’s test yielded a marginal result (β_1_ = −4.5, *p* = 0.0496), providing limited evidence of publication bias. Moreover, smaller studies might have exaggerated effect sizes, potentially distorting the overall effect size estimate. A trim-and-fill analysis was conducted to account for potential missing studies; however, no imputations were made due to insufficient evidence of significant publication bias or inadequate number of studies for reliable analysis. Consequently, the reported effect size and confidence interval remained unchanged regardless of whether imputed data were included (β = −0.003, 95% CI: −0.225 to 0.219).

#### 3.3.4. Evidence of Certainty

A GRADE assessment was conducted to evaluate the certainty of the evidence pertaining to childhood linear growth and stunting. The results of the assessment indicated that the certainty of the evidence for both outcomes was moderate (Table 5 and Table 6).

## 4. Discussion

The current systematic review and meta-analysis investigated the effect of rainfall and temperature variations on childhood linear growth and stunting among under-five children (0–59 months) across the global tropics. The narrative synthesis highlights the multifaceted and geographically specific nature of the relationship for both the weather proxy indicators in full tropical countries. While some studies, particularly in sub-Saharan Africa [59], suggest a potential negative impact of rising temperatures and drier conditions on child linear growth and stunting, others show contrasting or even contradictory results. For example, some studies in sub-Saharan Africa [59] and Uganda [36] suggested negative impacts of hotter and drier conditions or droughts, while others in the same regions found potential decreases in stunting with rising temperatures [51]. In this regard, hotter temperature, drier climates, and drought exacerbate child stunting through their adverse impact on food security, access to potable water, and overall health, resulting in malnutrition, a primary determinant of stunted growth [56,75,76].

With regard to the rainfall variations, the link with childhood linear growth in the tropics seems complex and geographically specific. Several factors, including timing, amount, location-specific diseases, societal factors like conflict, and agricultural practices, contribute to this intricate relationship. For example, a study done in Kenya indicated the positive association of rainfall with linear growth [61], suggesting that increased rainfall improves child growth, potentially through better agricultural yields and nutrition. However, excessive rainfall reduces hunger but can also lead to waterborne illnesses and hinder nutrition, as shown in Ghana [39]. Contrary to this, studies in Ethiopia suggest that less rain during dry seasons is beneficial [7], while overall higher rainfall can worsen stunting [35]. Apart from the amount of rainfall, increased variability in rainfall patterns was linked to a decline in children’s height-for-age z-scores, potentially indicating higher stunting rates [72]. However, several studies did not find a clear association between rainfall and stunting [41,42,43,56,77]. The discrepancies observed across studies may be attributable to a variety of factors. While several studies have identified potential correlates of improved child nutrition even during drought (i.e., low rainfall), due to agricultural diversification, crop production, and trade, the specific mechanisms by which these factors mitigate drought’s impact on child nutrition remain under-explored [78]. Moreover, a positive association between increased average monthly precipitation and reduced risk of childhood malnutrition has been established. The relationship is likely indirect, with changes in precipitation affecting child nutritional status over time through mechanisms such as altered water availability and subsequent impacts on crop yields and food security [79]. Conversely, as demonstrated by a study conducted in Ethiopia, exposure to drought is associated with increased vulnerability to child undernutrition. Household-level factors, including parental education and livelihood strategies, further modulate this relationship [80].

The separate meta-analysis in the current study investigated the broader association between rainfall, temperature, and childhood linear growth in tropical regions. It confirms the complexities of the weather metrics included. The analysis revealed a small and positive effect of rainfall on child growth, suggesting a potential benefit from increased rainfall or precipitation. It also highlighted a significant negative association between lower rainfall and child linear growth. Furthermore, the analysis found that higher temperatures are linked to decreased linear growth. Interestingly, a small positive association was found between temperature pattern and childhood stunting. These findings further emphasize the intricate and geographically specific nature of weather patterns’ influence on child growth in tropical contexts. While both rainfall and temperature patterns seem to play a role, a more comprehensive understanding necessitates further research into the complex interplay between weather, environmental factors, and socio-economic conditions.

Studies examining weather patterns (rainfall and temperature) suggest that they indirectly influence childhood malnutrition through complex routes, like impacting agricultural production [20,35,81,82]. Our meta-analysis revealed a weak overall effect. While the effect was positive (meaning some association between weather and malnutrition exists), the strength of that association was weak, with a value of 0.026. Overall, our meta-analysis provides a starting point for understanding the complex relationship between rainfall, temperature patterns, and child stunting. It is also important to note that many factors might contribute to the variations in child growth, greatly masking the effects of weather patterns. Numerous socio-economic and demographic factors moderate the correlation between weather variables and malnutrition in the tropical countries. These factors encompass age, socio-economic status, gender, and maternal nutrition. Age is a significant determinant of malnutrition outcomes, with children aged 1 to 2 years exhibiting the greatest vulnerability to the adverse effects of climate change on nutrition [22,83]. The poorest segments of the population are disproportionately susceptible to the negative impacts of climate change on nutrition. However, government or international food aid interventions can mitigate these effects [84]. Gender also plays a moderating role, with girls being more vulnerable than boys to malnutrition during droughts, while boys may be more susceptible to the adverse consequences of increased precipitation on nutrition [83]. Maternal education level can further influence this relationship, as higher levels of maternal education are associated with reduced risk of malnutrition in children [85].

## 5. Strengths and Limitations

This systematic review and meta-analysis provides a comprehensive summary of the effects of temperature and rainfall patterns on childhood linear growth. The review focused on studies utilizing national demographic and health surveys from tropical countries. The inclusion of large sample sizes in these surveys enhanced the precision and representativeness of the samples and relevance of the findings. Our systematic review and meta-analysis have some limitations related to geographic scope and weather variables that affect generalizability. Firstly, most studies originated in Africa. While Indonesia and Peru are included, other tropical regions in South America and Asia are not well represented. This limits the application of the findings to these areas. Studies from partially tropical countries were also excluded as they share the weather patterns of other non-tropical regions, so the results might not be applicable to those regions either. Secondly, the analysis only considered rainfall and temperature as indicators of weather. Other weather extremes, like floods or droughts, were not included. This might limit our understanding of the full impact of weather on child undernutrition. Finally, our meta-analysis investigating the impact of temperature patterns on child linear growth has limitations concerning the included effect sizes. The analysis incorporated effect sizes derived from single studies that examined extreme temperature levels. This approach might violate the assumption of effect size independence. Furthermore, the inclusion of a mean daily temperature effect size may not adequately capture the concept of temperature pattern. In terms of precision, these limitations could potentially lead to an overestimation of the overall effect size. However, it is believed that the meta-analysis provides useful insights as to the climate change elements and child nutrition and growth in the global tropics.

## 6. Implications of the Study

The current systematic review and meta-analysis investigated rainfall and temperature patterns as a potential contributor to childhood linear growth and chronic undernutrition in the tropics. However, the findings suggest a more intricate relationship. In the narrative synthesis, while increased rainfall can enhance food production in some areas, excessive precipitation can lead to disease outbreaks and hinder child growth. Temperature variations also present a twofold impact, particularly in the context of climate change. Hotter temperatures might negatively impact child growth; however, some studies suggest a potential decrease in stunting with rising temperatures. Thus policymakers in tropical countries experiencing significant temperature variability should prioritize agricultural adaptations and nutritional programs to buffer against the negative impacts on child growth. The narrative synthesis has also shown that specific rainfall and temperature patterns coupled with complex factors can either increase or decrease childhood linear growth and the risk of stunting. It also shows high heterogeneity of results across studies coupled with a lack of research in many countries in the Asian and South American continents, specifically in the full tropical region. Future research is required to at least highlight the mediating and moderating factors in the impact of the weather metrics on childhood linear growth.

Further, in our meta-analysis, a weak overall association between the weather metrics and childhood linear growth was found. This suggests that factors beyond weather likely play a significant role. These factors could include physiological conditions during pregnancy, other environmental conditions influencing disease incidence, socio-economic factors, agricultural practices, and social unrest or political instability. Our meta-analysis has also shown a beneficial effect of rainfall on linear growth, although the positive association may be mediated by various factors. For instance, in countries where agriculture is heavily dependent on rainfall, increased precipitation can lead to higher crop yields. The improved food security can positively impact childhood nutrition and growth. However, the extent to which these benefits are realized can vary depending on moderating factors such as location and socio-economic status. While urban residents may experience both increased agricultural yield and protection from excess rain, rural residents, especially those living in low-income farming communities, may only benefit from the agricultural yield. In this regard, actions are required to protect children from extreme weather conditions mainly related to rainfall and temperature in the rural areas. Several factors, including timing and amount of rainfall, location-specific diseases, societal factors like conflict, and agricultural practices, contribute to this intricate relationship. Children in lower- and middle-income countries are the most vulnerable to climate change resulting in malnutrition. This suggests that actions from policymakers are important in creating an enabling environment for partnership to at least mitigate the effects of extreme weather conditions across the countries in the tropics. Actions by the countries in the tropics are also required to align and strengthen their policies and developmental activities towards ensuring the Sustainable Development Goals (SDGs) of the United Nations, to avert future threats of climate change, specifically with the goals of no poverty, zero hunger, good health and well-being, quality education, gender equality, clean water and sanitation, peace, justice, and strong institutions; decent work and economic growth; sustainable cities and communities; responsible consumption and production; partnership in reaching these goals; and climate action.

## 7. Conclusions

This systematic review and meta-analysis identified studies across the full global tropics that link temperature and rainfall patterns, serving as a weather proxy, to chronic child undernutrition. Heterogeneous conclusions emerged in the original studies, with some suggesting that rainfall and temperature patterns can either increase or decrease the childhood linear growth and risk of stunting, while others found no clear association. Our meta-analysis showed an association of higher rainfall with an increase in linear growth, while low rainfall was associated with a significant decrease in linear growth and increased risk of stunting. Extreme temperatures and mean daily temperatures were associated with a decrease in linear growth. These findings emphasize how complex and location-specific the link is between weather patterns and childhood development in tropical environments. While both rainfall and temperature pattern appear to exert some influence on child growth, a comprehensive understanding of this association necessitates further exploration of the complex interplay between weather, other environmental factors, and socio-economic conditions. The implementation of climate change adaptation strategies, such as sustainable agriculture and water irrigation practices, coupled with enhanced nutritional interventions targeted at children, is imperative. Additionally, by improving access to educational resources and basic health infrastructure, the rise in child malnutrition could be mitigated. Policymakers must actively create a conducive environment for partnerships to mitigate the adverse effects of extreme weather events in tropical regions. Additionally, tropical countries should harmonize and strengthen their policies and development initiatives to align with the United Nations Sustainable Development Goals (SDGs). This alignment is crucial to prevent future climate-change-related threats.

## Figures and Tables

**Figure 1 ijerph-21-01269-f001:**
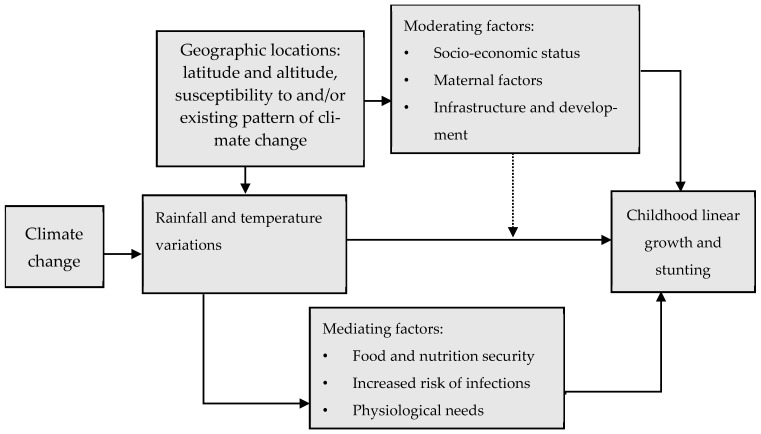
Conceptual framework showing how climate-change-induced temperature and rainfall variations affect childhood linear growth. Source: Authors’ own elaboration.

**Figure 2 ijerph-21-01269-f002:**
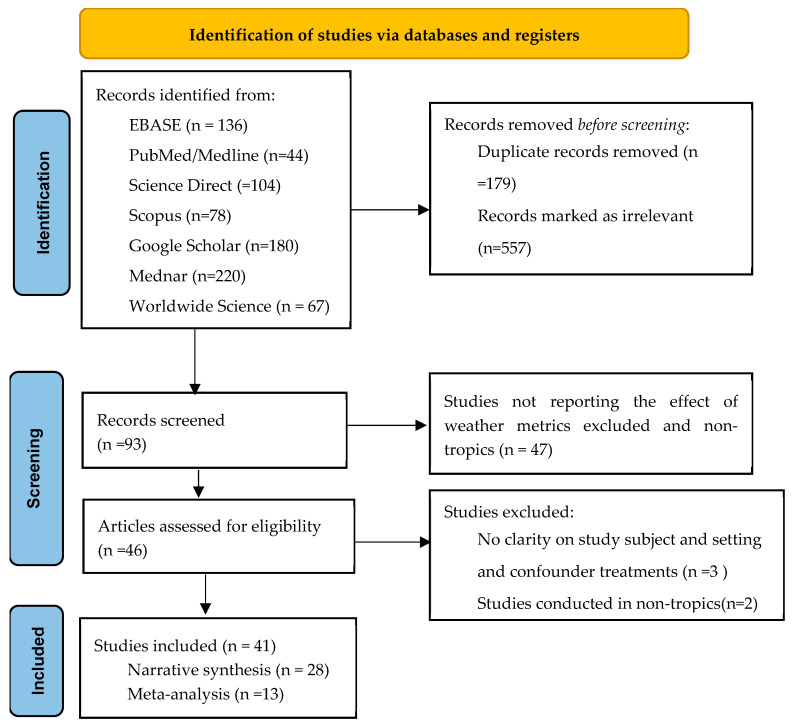
PRISMA flow diagram illustrating the number of studies included in the systematic review and meta-analysis.

**Figure 3 ijerph-21-01269-f003:**
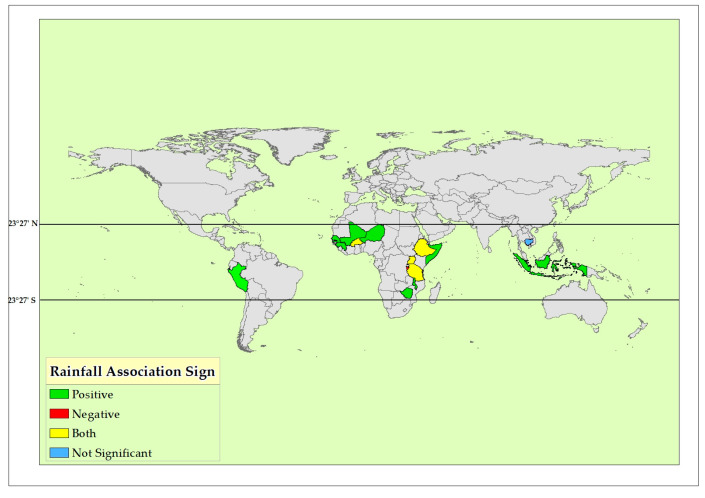
Effect of rainfall patterns on childhood linear growth reported by the original studies in the global tropics (2024).

**Figure 4 ijerph-21-01269-f004:**
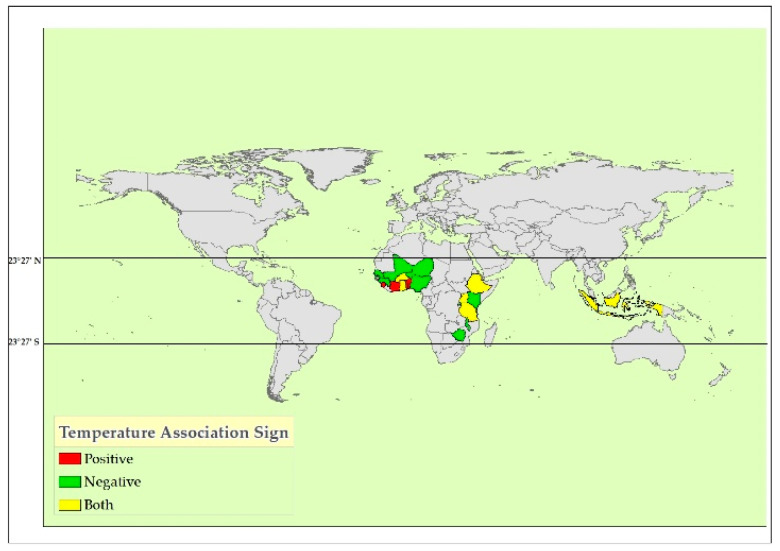
Effect of temperature pattern on childhood linear growth reported by the original studies in the global tropics (2024).

**Figure 5 ijerph-21-01269-f005:**
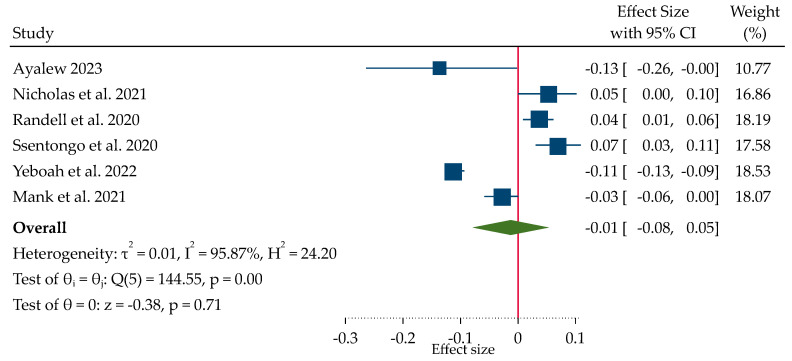
Forest plot of pooled effect size (β) showing the effect of rainfall pattern on childhood linear growth (height-for-age z-score) [7,64,69,70,71,72].

**Figure 6 ijerph-21-01269-f006:**
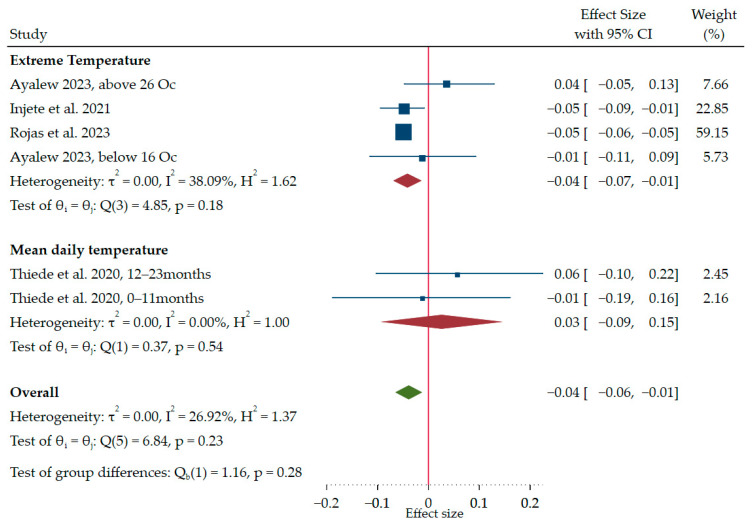
Forest plot of pooled effect size (β) showing the effects of temperature patterns on childhood linear growth [41,64,66,68].

**Figure 7 ijerph-21-01269-f007:**
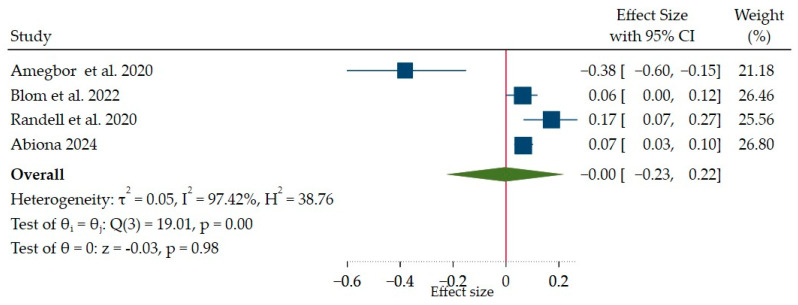
Forest plot of pooled effect size (β) showing the effects of temperature patterns on childhood linear growth failure (height-for-age z-score < −2) [7,36,65,67].

**Table 3 ijerph-21-01269-t003:** Subgroup analysis pooled standardized regression coefficients (β) showing the effects of rainfall patterns on childhood linear growth (height-for-age z-score).

Group	No. of Studies	Effect Size	95% Confidence Interval	*p*-Value	Heterogeneity Statistics				
					Q	P > Q	τ^2^	% I^2^	H^2^
Rainfall pattern									
Higher rainfall	3	0.049	0.024, 0.073	0.000	2.14	0.344	0.000	25.50	1.34
Lower rainfall	3	−0.080	−0.140, −0.020	0.009	22.59	0.000	0.002	87.88	8.25
Region									
East Africa	3	0.005	−0.107, 0.118	0.924	9.04	0.011	0.009	94.14	17.06
West Africa	2	−0.070	−0.148, 0.007	0.076	22.20	0.000	0.003	95.32	21.36
Sample size									
≤10,000	2	0.020	−0.075, 0.114	0.685	15.68	0.000	0.004	93.29	14.91
≥10,001	4	−0.032	−0.123, 0.059	0.495	104.29	0.000	0.008	96.36	27.50

**Table 4 ijerph-21-01269-t004:** Subgroup analysis of pooled standardized regression coefficients (β) showing the effects of temperature patterns on childhood linear growth failure (HAZ < −2SD).

Group	No. of Studies	Effect Size	95% Confidence Interval	*p*-Value	Heterogeneity Statistics				
					Q	P > Q	τ^2^	% I^2^	H^2^
Region									
East Africa	2	−0.094	−0.614, 0.427	0.724	18.9	0.000	0.134	94.47	18.09
West Africa	2	0.064	0.035, 0.093	<0.0001	0.04	0.839	0.000	0.08	1.00
Sample size									
≤10,000	2	−0.139	−0.560, 0.281	0.515	14.64	0.000	0.086	92.8	13.88
≥10,001	2	0.105	0.004, 0.207	0.042	3.39	0.066	0.004	68.06	3.13

**Table 5 ijerph-21-01269-t005:** GRADE evidence for the effects of rainfall on childhood linear growth (height-for-age z-score).

No. of Studies	Study Design	Certainty Assessment				Sample Size	Certainty	Strength of Recommendation
		Risk of Bias	Inconsistency	Indirectness	Imprecision			
6	Observational studies	Not serious	Serious ^a^	Not serious	Not serious	73,922	Moderate	Conditional

^a^ Serious inconsistency due to high heterogeneity (I^2^ = 95.87%) that was largely unexplained in pre-specified subgroup and sensitivity analyses.

**Table 6 ijerph-21-01269-t006:** GRADE evidence for the effects of temperature on childhood linear growth (height-for-age z-score) and stunting.

No. of Studies	Study Design	Certainty Assessment				Sample Size	Certainty	Strength of Recommendation
		Risk of Bias	Inconsistency	Indirectness	Imprecision			
Linear growth								
4	Observational studies	Not serious	Not serious	Not serious	Not serious	42,537	Moderate	Conditional
Stunting								
4	Observational studies	Not serious	Serious ^b^	Not serious	Not serious	60,364	Moderate	Conditional

^b^ Serious inconsistency due to high heterogeneity (I^2^ = 97.42%) that was largely unexplained in pre-specified subgroup and sensitivity analyses.

## Data Availability

The individual datasets used are not publicly available due to confidentiality agreements with the original study authors. However, summary statistics and effect sizes are provided in the Section 3. Requests for access to the datasets can be made to the corresponding author.

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
