# Peer review of "The Effect of Rainfall and Temperature Patterns on Childhood Linear Growth in the Tropics: Systematic Review and Meta-Analysis"

_ijerph, 2024, doi:10.3390/ijerph21101269_

Round 1
Reviewer 1 Report
Comments and Suggestions for Authors
Dear Authors, please find my detailed feedback that should help you develop and improve your text for the upcoming revision.
Abstract
According to the abstract, “An increase in every one standard deviation of rainfall results in a 0.049 standard deviation increase in linear growth.” Given that the effects of climate change can differ significantly by location, this statement might benefit from clarification regarding the circumstances or context in which this link is true. If appropriate, advise the writers to briefly discuss any demographic or regional modifiers.
There is unclear reporting of the effect magnitude for temperature variability. If non-linear effects or thresholds are found in addition to linear connections, this should be made clear in the abstract. Put this differently: “Temperature variability showed a negative correlation with linear growth, where each standard deviation increase resulted in a decrease in growth by 0.039 standard deviations, with specific impacts varying by regional climates.”
The limitations of the study's scope, especially with regard to the geographical coverage and the types of studies included (e.g., observational vs. experimental), may be more clearly highlighted in the abstract. Suggest including a remark along the lines of, “While our findings provide valuable insights, they are primarily based on observational studies from sub-Saharan Africa and may not be generalizable to other tropical regions.”
Introduction
A more thorough review of earlier meta-analyses on rainfall, temperature variability, and child growth could improve the introduction. It should specifically discuss how this meta-analysis closes a gap by concentrating on children under five, a demographic that is most vulnerable. Include the following suggestion: “Previous reviews have focused broadly on undernutrition without differentiating impacts by age groups. Our study contributes to the literature by honing in on under-fives, who are at a critical developmental stage.”
A more precise theoretical framework connecting climatic variability to nutritional effects would improve the text. A thorough explanation of how temperature and rainfall directly impact agricultural outputs and food availability, which in turn affect childhood growth, is absent from the current draft. Suggest expanding on models or hypotheses that already exist in this field.
Methods
A thorough explanation of the methodology employed in the studies that made up the meta-analysis is absent from the manuscript. The authors should specifically address the variety of methodological methods, including the statistical models employed, the ways in which confounders were accounted for, and the effects of these methodological variations on the study's conclusions, in order to improve the validity and application of the findings.
The systematic review's description of the databases and search phrases is imprecise. Include precise search terms, search dates, and any filters or language limitations used for robustness. The following search string was used across all databases, "[Exact Search String]," with filters to include only peer-reviewed English-language publications published between 2000 and 2024.
The text has to explain the inclusion/exclusion of specific study designs. If only studies with height-for-age Z-scores were included, explain the exclusion of other growth indicators, as this may have an impact on the analysis's comprehensiveness.
The text has to explain the inclusion/exclusion of specific study designs. If only studies with height-for-age Z-scores were included, explain the exclusion of other growth indicators, as this may have an impact on the analysis's comprehensiveness.
Results
The results list a number of subgroup analyses, however they don't go into enough detail to explain how or why these subgroups are important. If the analysis is done geographically, for instance, the article should explain the selection of various places (e.g., distinct climate patterns, socioeconomic conditions).
There are studies in the meta-analysis that present conflicting results. Further exploration of the possible causes of these disparities, such as variations in research design, population demography, or regional environmental circumstances, is warranted. Include the following suggestion: "The conflicting results across studies may reflect underlying variations in regional dietary diversity, agricultural practices, and community resilience to climate impacts."
Discussion
While the discussion touches on the implications of the findings, it lacks specific recommendations for policymakers or practitioners. Suggest including targeted advice for specific stakeholder actions based on the results, such as: “Policymakers in regions experiencing significant temperature variability should prioritize agricultural adaptations and nutritional programs to buffer the negative impacts on child growth.”
The study's overly expansive definition of temperature and rainfall variability raises the possibility of inconsistent interpretations of the impact on child growth. It is essential to provide a more precise description for these terms or to clarify how the definitions from other research were used for the meta-analysis.
There is a disparity in the study's coverage of tropical regions, with a strong emphasis on sub-Saharan Africa. To increase the generalizability of the results, this regional concentration should be clearly justified or more research from other tropical locations, such as Southeast Asia and Latin America, should be included.
While the manuscript mentions the use of forest plots and I-squared statistics, there is no detailed discussion of how heterogeneity among the studies was handled beyond these measures. A more thorough analysis, perhaps including a sensitivity analysis or meta-regression, could provide deeper insights into the sources of heterogeneity.
Although noted, the effect of publication bias on the study's findings is not sufficiently addressed. A more thorough publication bias analysis, utilizing tools like a funnel plot and Egger's test, must to be carried out and reported by the authors. They should also go over how missing studies can affect the study's findings.
A more thorough analysis of the socioeconomic variables that moderate the link between climatic variability and child growth would be beneficial for the publication. It should specifically investigate the ways in which varying degrees of socioeconomic status, health infrastructure, and community resilience affect the results that are seen.
A critical assessment of the caliber of the evidence from the included research is absent from the discussion section. In order to give readers a better understanding of the strength of the data, the authors should evaluate the relationships between climate variability and child growth. They could consider use a grading system similar to GRADE.
Only a cursory mention of the implications for future research is made. The text should identify specific areas that require additional research, especially in determining the causal relationships between children's nutritional outcomes and climate variability.
Conclusion
To emphasize the wider significance and urgency of the research, the manuscript's conclusion might be reinforced by presenting the major findings in the context of global health goals, such as the Sustainable Development Goals (SDGs).
The possibilities for adaptation and intervention measures that could lessen the detrimental effects of climatic variability on child growth are not properly covered in the current study. To give stakeholders practical insights, it would be helpful to include a discussion of potential adaptation measures, including improved farming techniques or community-based nutrition initiatives.
Comments on the Quality of English LanguageThe manuscript would benefit from a thorough review of the English language to enhance clarity and coherence; specifically, attention should be paid to grammatical accuracy, consistency in terminology, and the simplification of complex sentences to improve overall readability.
Author Response
|
Dear Reviewer, Thank you very much for taking the time to review this manuscript. Please find the detailed responses below and the corresponding revisions and corrections highlighted in the re-submitted files. |
||
|
Abstract |
|
|
|
Comments 1: According to the abstract, “An increase in every one standard deviation of rainfall results in a 0.049 standard deviation increase in linear growth.” Given that the effects of climate change can differ significantly by location, this statement might benefit from clarification regarding the circumstances or context in which this link is true. If appropriate, advise the writers to briefly discuss any demographic or regional modifiers. |
||
|
Response 1: Thank you for your valuable feedback. We agree with your comment and have revised our analysis accordingly stating. The change can be found in line number 24-28 of page 1. |
||
|
Comments 2: There is unclear reporting of the effect magnitude for temperature variability. If non-linear effects or thresholds are found in addition to linear connections, this should be made clear in the abstract. Put this differently: “Temperature variability showed a negative correlation with linear growth, where each standard deviation increase resulted in a decrease in growth by 0.039 standard deviations, with specific impacts varying by regional climates.” |
||
|
Response 2: Thank you for your valuable feedback. We appreciate your suggestion and have incorporated it into our analysis. As recommended, we have differently stated. The change can be found in line number 29-31 of page 1. Comments 3 The limitations of the study's scope, especially with regard to the geographical coverage and the types of studies included (e.g., observational vs. experimental), may be more clearly highlighted in the abstract. Suggest including a remark along the lines of, “While our findings provide valuable insights, they are primarily based on observational studies from sub-Saharan Africa and may not be generalizable to other tropical regions.” Response 3: Thank you for your valuable feedback. We appreciate your suggestion and have incorporated it into our analysis. As recommended, we have highlighted in the abstract. The change can be found in line number 35-39 of page 1. Introduction Comments 4: A more thorough review of earlier meta-analyses on rainfall, temperature variability, and child growth could improve the introduction. It should specifically discuss how this meta-analysis closes a gap by concentrating on children under five, a demographic that is most vulnerable. Include the following suggestion: “Previous reviews have focused broadly on undernutrition without differentiating impacts by age groups. Our study contributes to the literature by honing in on under-fives, who are at a critical developmental stage.” Response 4: Thank you for your valuable feedback. We agree with your comment and have thoroughly reviewed and addressed the existing literature gap. As suggested, we have reviewed earlier meta-analysis and included in our text. The change can be found in line number 98-107 of page 3. Comments 5: A more precise theoretical framework connecting climatic variability to nutritional effects would improve the text. A thorough explanation of how temperature and rainfall directly impact agricultural outputs and food availability, which in turn affect childhood growth, is absent from the current draft. Suggest expanding on models or hypotheses that already exist in this field. Response 5: Thank you for your valuable feedback. We agree with your comment and have incorporated how agricultural outputs and food availability are impacted and finally affecting child growth. The change can be found in line number 85-89 of page 2. Methods Comments 6: A thorough explanation of the methodology employed in the studies that made up the meta-analysis is absent from the manuscript. The authors should specifically address the variety of methodological methods, including the statistical models employed, the ways in which confounders were accounted for, and the effects of these methodological variations on the study's conclusions, in order to improve the validity and application of the findings. Response 6: Thank you for your valuable feedback. We agree with your comment and have modified specifying the methodologies employed in each of the included studies in our meta-analysis. The explanations are specified in the methods and materials and result sections of our review. The change can be found in line number 182-190 of page 5, and Table 2 of page 6-7. Comments 7: The systematic review's description of the databases and search phrases is imprecise. Include precise search terms, search dates, and any filters or language limitations used for robustness. The following search string was used across all databases, "[Exact Search String]," with filters to include only peer-reviewed English-language publications published between 2000 and 2024. Response 7: Thank you for your valuable feedback. We agree with your comment and have ensured that the search phrases are precisely stated, taking into account the initial strategies employed. Only the peer-reviewed and English language publications were included. However, during the grey literature search the non-peer-reviewed papers were also searched not to miss them. The change can be found in line number 122-135 of page 3-4. Comments 8: The text has to explain the inclusion/exclusion of specific study designs. If only studies with height-for-age Z-scores were included, explain the exclusion of other growth indicators, as this may have an impact on the analysis's comprehensiveness. Response 8: Thank you for your valuable feedback. We agree with your comment and have already provided the explanation of the inclusion and exclusion criteria for study designs. Only studies that presented data on height-for-age Z-scores and stunting were included in our analysis. The explanation was already found in line number 138-157 of page 4. Comments 9: The text has to explain the inclusion/exclusion of specific study designs. If only studies with height-for-age Z-scores were included, explain the exclusion of other growth indicators, as this may have an impact on the analysis's comprehensiveness. Response 9: Similar with comments 8 above. Results Comments 10: The results list a number of subgroup analyses, however they don't go into enough detail to explain how or why these subgroups are important. If the analysis is done geographically, for instance, the article should explain the selection of various places (e.g., distinct climate patterns, socioeconomic conditions). Response 10: Thank you for your valuable feedback. We agree with your comment and have incorporated a detailed explanation of why subgroup analysis was required. The change can be found in line number 432-439 of page 14. Comments 11: There are studies in the meta-analysis that present conflicting results. Further exploration of the possible causes of these disparities, such as variations in research design, population demography, or regional environmental circumstances, is warranted. Include the following suggestion: "The conflicting results across studies may reflect underlying variations in regional dietary diversity, agricultural practices, and community resilience to climate impacts." Response 11: Thank you for your valuable feedback. We agree with your comment and have presented possible causes of disparities, incorporating the suggested statement. The change can be found in line number 399-403 of page 12. Discussion Comments 12: While the discussion touches on the implications of the findings, it lacks specific recommendations for policymakers or practitioners. Suggest including targeted advice for specific stakeholder actions based on the results, such as: “Policymakers in regions experiencing significant temperature variability should prioritize agricultural adaptations and nutritional programs to buffer the negative impacts on child growth.” Response 12: Thank you for your valuable feedback. We agree with your comment and have included specific recommendations including the suggested text. The change can be found in line number 658-695 of page 20-21. Comments 13: The study's overly expansive definition of temperature and rainfall variability raises the possibility of inconsistent interpretations of the impact on child growth. It is essential to provide a more precise description for these terms or to clarify how the definitions from other research were used for the meta-analysis. Response 13: Thank you for your valuable feedback. We agree with your comment and have ensured that the terms are consistently defined throughout the manuscript, providing clear and concise explanations. The changes of temperature and rainfall variability to temperature and rainfall patterns has been made throughout the text. Comments 14: There is a disparity in the study's coverage of tropical regions, with a strong emphasis on sub-Saharan Africa. To increase the generalizability of the results, this regional concentration should be clearly justified or more research from other tropical locations, such as Southeast Asia and Latin America, should be included. Response 14: Thank you for your valuable feedback. We agree with your comment and have explicitly stated the limitations related to the coverage of our study in the limitations section. The limitation related to study coverage was already found in line number 632-638 of page 20. Comments 15: While the manuscript mentions the use of forest plots and I-squared statistics, there is no detailed discussion of how heterogeneity among the studies was handled beyond these measures. A more thorough analysis, perhaps including a sensitivity analysis or meta-regression, could provide deeper insights into the sources of heterogeneity. Response 15: Thank you for your valuable feedback. We agree with your comment and have addressed how heterogeneity was handled in our analysis. Sensitivity analysis and the identification of sources of heterogeneity were included in the results section. To effectively handle heterogeneity, we conducted sensitivity analysis, subgroup analysis to identify sources of heterogeneity, employed a random-effects model, and examined publication biases. These procedures were detailed in the materials and methods (line number 235-250, page 6) and reported in the meta-analysis section of the results (line number 432-439, page 14; line number 488-498, page 16; line number 532-542, page 17). Comments 16: Although noted, the effect of publication bias on the study's findings is not sufficiently addressed. A more thorough publication bias analysis, utilizing tools like a funnel plot and Egger's test, must to be carried out and reported by the authors. They should also go over how missing studies can affect the study's findings. Response 16: Thank you for your valuable feedback. We agree with your comment and have sufficiently addressed the effect of publication bias. All procedures conducted to identify the effect of publication bias are reported in the results section. The detailed publication bias reports are found in line number 432-439, page 14; line number 488-498, page 16; line number 532-542, page 17). The funnel plots are included in the Supplementary S3-4 of supplementary material of the manuscript. Comments 17: A more thorough analysis of the socioeconomic variables that moderate the link between climatic variability and child growth would be beneficial for the publication. It should specifically investigate the ways in which varying degrees of socioeconomic status, health infrastructure, and community resilience affect the results that are seen. Response 17: Thank you for your valuable feedback. We agree with your comment and have included how socioeconomic variables moderate the link between weather variables and child growth. The analysis and explanation with corresponding references can be found in line number 615-627 of page 19-20. Comments 18: A critical assessment of the caliber of the evidence from the included research is absent from the discussion section. In order to give readers a better understanding of the strength of the data, the authors should evaluate the relationships between climate variability and child growth. They could consider use a grading system similar to GRADE. Response 18: Thank you for your valuable feedback. We agree with your comment and have included the assessment at the end of the materials and result section. The GRADE evidence can be found in materials and methods section (line number 216-220 of page 5) and result section (line number 543-554 of page 18). Comments 19: Only a cursory mention of the implications for future research is made. The text should identify specific areas that require additional research, especially in determining the causal relationships between children's nutritional outcomes and climate variability. Response 19: Thank you for your valuable feedback. We agree with your comment and have included specific areas requiring additional research in several section of the manuscript. The identified areas can be found in Strength and Limitations (line number 632-641 of page 20), Implications of the Study (line number 665-667 of page 20) and Conclusion (line number 707- 710 of page 21). Conclusion Comments 20: To emphasize the wider significance and urgency of the research, the manuscript's conclusion might be reinforced by presenting the major findings in the context of global health goals, such as the Sustainable Development Goals (SDGs). Response 20: Thank you for your valuable feedback. We agree with your comment and have reinforced the presentation of the conclusion within the context of SDGs. The change can be found in line number 715-718 of page 21. Comments 21: The possibilities for adaptation and intervention measures that could lessen the detrimental effects of climatic variability on child growth are not properly covered in the current study. To give stakeholders practical insights, it would be helpful to include a discussion of potential adaptation measures, including improved farming techniques or community-based nutrition initiatives. Response 21: Thank you for your valuable feedback. We agree with your comment and have included specific practical insights about possible adaptation and intervention measures. The possible measures. The included potential measures can be found in line number 710-718 of page 21. |
||
|
4. Response to Comments on the Quality of English Language |
||
|
Point 1: The manuscript would benefit from a thorough review of the English language to enhance clarity and coherence; specifically, attention should be paid to grammatical accuracy, consistency in terminology, and the simplification of complex sentences to improve overall readability. |
||
|
Response 1: Thank you for your valuable feedback. We agree with your comment and have revised the manuscript to enhance its overall quality and readability. We have specifically focused on improving the terminology and language usage. |
||

Reviewer 2 Report
Comments and Suggestions for Authors
Comments to Systematic Review-Manuscript ID: ijerph-3170477
Recommended for a moderate revision!
The manuscript presents a very elaborative study with a focus on systematic review and meta-analysis on the effect of rainfall and temperature variability on childhood linear growth among under-fives in the tropics. I would suggest to justify/incorporate the below given comments accordingly, for general and related researchers. That would make this paper interesting to readers and make it sort more novel one. I suggest please looking into below.
Some moderate comments:
In abstract section + justification for retrieved results: on line 23 – 27: “Whereas temperature variability was associated with a reduced linear growth with β = -0.039, 95% CI: -25 0.065 to -0.013. Additionally, our meta-analysis shows a small but positive relationship of childhood stunting with temperature variability in the western Africa (β = 0.064, 95% CI: 27 0.035, 0.093).”
Page 1, in introduction section: Possibly elaborate/add some literature about the correlative factor of variability that might have potentially impacted the nutritional imbalance in food crops/food of tropics, particularly in perspective of food security.
Page 4, on lines 131 “2.3 Study variables”: Possibly include/mention any strong justification for selecting the temperature and rainfall as study variables, any relation in context of food security???
Page 5, in results and discussion section: It would be better to mention the cross-correlation (with graphical representation or info graph) regarding the cross-results/inverse results retrieved from temperature and precipitation on child linear growth or stunting
If possible, to show it also in mapping if feasible to export from any such deta/meta-data in GIS-environment
Author Response
|
Dear Reviewer, Thank you very much for taking the time to review this manuscript. Please find the detailed responses below and the corresponding revisions/corrections highlighted in the re-submitted files and point-by-point response below. |
||||
|
Comments 1: In abstract section + justification for retrieved results: on line 23 – 27: “Whereas temperature variability was associated with a reduced linear growth with β = -0.039, 95% CI: -25 0.065 to -0.013. Additionally, our meta-analysis shows a small but positive relationship of childhood stunting with temperature variability in the western Africa (β = 0.064, 95% CI: 27 0.035, 0.093).” |
||||
|
Response 1: Thank you for your valuable feedback. We agree with your comment and have provided a detailed explanation for the possible association between temperature variability and linear growth in the abstract. The justification can be found in line number 34-35 of page 1. |
||||
|
Comments 2: Page 1, in introduction section: Possibly elaborate/add some literature about the correlative factor of variability that might have potentially impacted the nutritional imbalance in food crops/food of tropics, particularly in perspective of food security. |
||||
|
Response 2: Thank you for your valuable feedback. We agree with your comment and have incorporated additional literature related to food security to enhance our in the introduction. The elaborations starts from line number 55 to 97 of page 2. Specifically, additional elaboration is added in line number 85-89 of page 2. Comments 3: Page 4, on lines 131 “2.3 Study variables”: Possibly include/mention any strong justification for selecting the temperature and rainfall as study variables, any relation in context of food security??? Response 3: Thank you for your valuable feedback. We agree with your comment and have incorporated. The changes can be found in line number 163-167 of page 4. Comments 4: Page 5, in results and discussion section: It would be better to mention the cross-correlation (with graphical representation or info graph) regarding the cross-results/inverse results retrieved from temperature and precipitation on child linear growth or stunting Response 4: Thank you for your valuable feedback. We agree with your comment and have already included maps illustrating the association between weather variables and child linear growth in Figures 3 (page 8) and 4 (page 10). These maps provide a visual representation of the geographical variations in these relationships. |

Round 2
Reviewer 1 Report
Comments and Suggestions for Authors
The revisions made are satisfactory.